# A Portable Nanoprobe for Rapid and Sensitive Detection of SARS-CoV-2 S1 Protein

**DOI:** 10.3390/bios12040232

**Published:** 2022-04-11

**Authors:** Hani A. Alhadrami, Ghadeer A. R. Y. Suaifan, Mohammed M. Zourob

**Affiliations:** 1Department of Medical Laboratory Technology, Faculty of Applied Medical Sciences, King Abdulaziz University, P.O. Box 80402, Jeddah 21589, Saudi Arabia; hanialhadrami@kau.edu.sa; 2Molecular Diagnostic Laboratory, King Abdulaziz University Hospital, King Abdulaziz University, P.O. Box 80402, Jeddah 21589, Saudi Arabia; 3Department of Pharmaceutical Sciences, Faculty of Pharmacy, The University of Jordan, Amman 11942, Jordan; gh.suaifan@ju.edu.jo; 4Department of Chemistry, Alfaisal University, Al Zahrawi Street, Al Maather, Al Takhassusi Rd., Riyadh 11533, Saudi Arabia

**Keywords:** SARS-CoV-2, COVID-19, rapid biosensors, onsite detection, low-cost diagnostic assay

## Abstract

Simple, timely, and precise detection of SARS-CoV-2 in clinical samples and contaminated surfaces aids in lowering attendant morbidity/mortality related to this infectious virus. Currently applied diagnostic techniques depend on a timely laboratory report following PCR testing. However, the application of these tests is associated with inherent shortcomings due to the need for trained personnel, long-time centralized laboratories, and expensive instruments. Therefore, there is an interest in developing biosensing diagnostic frontiers that can help in eliminating these shortcomings with a relatively economical, easy-to-use, well-timed, precise and sensitive technology. This study reports the development of fabricated Q-tips designed to qualitatively and semi-quantitatively detect SARS-CoV-2 in clinical samples and contaminated non-absorbable surfaces. This colorimetric sensor is engineered to sandwich SARS-CoV-2 spike protein between the lactoferrin general capturing agent and the complementary ACE2-labeled receptor. The ACE2 receptor is decorated with an orange-colored polymeric nanoparticle to generate an optical visual signal upon pairing with the SARS-CoV-2 spike protein. This colorimetric change of the Q-tip testing zone from white to orange confirms a positive result. The visual detection limit of the COVID-19 engineered colorimetric Q-tip sensor was 100 pfu/mL within a relatively short turnaround time of 5 min. The linear working range of quantitation was 10^3^–10^8^ pfu/mL. The engineered sensor selectively targeted SARS-CoV-2 spike protein and did not bind to another coronavirus such as MERS-CoV, Flu A, or Flu B present on the contaminated surface. This novel detection tool is relatively cheap to produce and suitable for onsite detection of COVID-19 infection.

## 1. Introduction

The novel coronavirus SARS-CoV-2 was announced as an emerging global pandemic on 11 March 2020 [1]. Recent clustering studies confirmed the spread of the SARS-CoV-2 virus from human to human [2]. The most reported symptoms of COVID-19 infection vary from mild illness, to severe pneumonia, to severe respiratory syndrome and multi-organ failure [3]. Months ago, the FDA approved several laboratory tests developed by the biggest diagnostic companies for the detection of SARS-CoV-2 and the diagnosis of COVID-19 infection. These laboratory tests include reverse transcriptase-polymerase chain reaction (RT-PCR) that detects SARS-CoV-2 RNA and a serological test that detects IgM and IgG antibodies produced in response to infection. SARS-CoV-2 IgM antibody detection provides a hint of recent exposure, whereas SARS-CoV-2 IgG antibody detection indicates a long period of infection [4]. Negative results reported from these tests do not eliminate the infection possibility of COVID-19 infection and should not be relied on as the sole basis for patient treatment and management decisions. Furthermore, cross-reactivity with other viruses and bacterial infections should be taken into consideration while reporting positive results of COVID-19 infection. The current tests used for the detection of COVID-19 infection are expensive and require trained biomedical laboratory personnel for conducting real-time PCR and other molecular techniques. Thus, a portable, cheap, accurate, and rapid diagnostic test to quickly identify infected patients and asymptomatic carriers to prevent virus transmission and assure timely patient treatment becomes more interesting day by day. Scientists’ efforts are oriented toward the characterization of potential molecular targets, pivotal for the detection of SARS-CoV-2 [5,6] and the development of vaccines and therapies.

In parallel with PCR and ELISA, several biosensors have been developed for the detection and diagnosis of SARS-CoV-2. This is due to the attractive features of the biosensors, such as high sensitivity, selectivity, and short analysis time. In general, biosensor platforms are based on three important elements: the identification molecular target (antigens, virus, spike protein [7]), antibody (IgG, IgM), protease (Mpro and PLp [8,9]), ACE-2, genetic RNA [10], the recognition element (based on antibodies, peptides, proteins, aptamers, and nucleic acid probes), and transduction and signal amplification systems (based on optical, electrochemical, and surface plasmon resonance; mechanical systems; and fluorescent signals) [11].

In March 2020, Zhu et al. reported the development of one-step reverse transcription loop-mediated isothermal amplification (RT-LAMP) with a nanoparticle-based biosensor (NBS). Although this biosensing system requires simple equipment, the system efficiency is temperature-sensitive, wherein the sample must be maintained at a constant temperature (63 °C) for 40 min during the test protocol. Furthermore, this assay is not suitable for onsite application, and the analysis procedure is completed in one hour.

Remarkably, protein-based nanobiotechnology plays an essential role in the clinical application of diagnostic biosensors [12,13]. The angiotensin-converting enzyme 2 (ACE2) protein receptor has been identified as the functional host receptor for SARS-CoV-2 spike protein. The ACE2 receptor mediates the virus entry and affects the pathophysiological processes [14].

Colorimetric biosensors are ideal candidates for point-of-care diagnostics of SARS-CoV-2 [15] because humans can easily identify the color change visually, so SARS-CoV-2 presence can be detected in symptomatic, asymptomatic, or paucysymptomatic cases at the early stages of the infection to prevent its transmission [16]. Moreover, the change in color can also be detected by a simple camera or intensity-based optical detector with relatively simple algorithms to quantify test results in a low-cost and speedy way. Furthermore, colorimetric biosensors do not require expensive analytical instrumentation [17,18,19,20]. In this context, several studies have been conducted in terms of appropriate colorimetric biosensor developments [21,22,23]. Hence, few rapid tests have been developed and used for this purpose [24]. However, the highly insufficient sensitivity value in these rapid antigen–antibody-based tests lowers their reliability [19,24]. As reported by Corman et.al., the sensitivity range of most investigated rapid tests overlaps with viral load figures from the infectious period in most patients. Accordingly, these assays could potentially be used in efforts to limit transmission but might not have the power to exclude SARS-CoV-2 infection in the very early and later phases of COVID-19 infection [25].

Herein, we report the development of a novel colorimetric-based biosensor for the rapid (5 min) and sensitive (100 pfu/mL) detection of SARS-CoV-2, based on the use of viral S1 spike protein as an identifying target. In this study, we illustrate that the binding of SARS-CoV-2 S1 protein with the complementary ACE2 receptor labeled with orange-colored polymeric nanoparticles resulted in a significant optical change in the Q-tip sensing zone color comparatively with viral concentrations (Figure 1).

## 2. Experimental

### 2.1. Materials and Reagents

1-ethyl-3-(3-dimethylaminopropyl) carbodiimide hydrochloride (EDC) and N-hydroxysuccinimide (NHS) were purchased from Fisher Scientific (Gillingham, Dorset, UK). Lactoferrin molecules were purchased from Monojo (Amman, Jordan). ACE2 human protein was purchased from Sino Biological (Beijing, China). Carboxylic functionalized, orange-colored polymers were obtained from Polysciences Inc. (Polysciences Inc., Warrington, PA, USA). The orange beads were 200 nm in size and packaged as 2.5% aqueous suspension (5.69 × 10^12^ beads/mL).

The activation step of the carboxylic functionalized polymeric nanobeads was performed with a mixture of EDC/NHS in phosphate buffer saline (PBS) buffer, pH 5.5. The nanoparticles were blocked using 1% bovine serum albumin (BSA) protein in PBS buffer, pH 7.4, to block the free carboxylic acid on the magnetic nanoparticles. All the aqueous solutions were prepared in Milli-Q-grade water.

### 2.2. Cell Line and SARS-CoV-2 Propagation

SARS-CoV-2 was isolated and propagated as described by Alhadrami et al. [26]. In brief, Dulbecco’s Eagle medium (DMEM) was used to grow and maintain Vero E6 cells (ATCC^®^ number 1568). A human nasopharyngeal swab with confirmed positive SARS-CoV-2 by RT-PCR result was used to isolate SARS-CoV-2 with gene accession number MT630432.1. Vero E6 cells were inoculated with 95% of the isolated SARS-CoV-2 and incubated with shaking every 15 min for 1 h at 37 °C in a CO_2_ incubator. Subsequently, the inoculum was replaced with 25 mL of viral inoculation medium, then incubated for 72 h at 37 °C in a CO_2_ incubator until a cytopathic effect (CPE) was observed. Ultimately, the supernatant was harvested and SARS-CoV-2 was stored at −80 °C. The plaque assay was conducted as described in the following section to determine the SARS-CoV-2 titer and TCID50. MERS-CoV was isolated from a human nasopharyngeal swab confirmed positive by RT-PCR. The MERS-CoV-positive sample was inoculated on the 95% confluent Vero E6 cells and finally, the virus was harvested as described above in the propagation of SARS-CoV-2.

### 2.3. Plaque Assay

The procedure for the plaque was followed as described by Alhadrami et al. [26]. Vero E6 cells were maintained in a DMEM medium and grown to 1 × 10^5^ mL^−1^. Two mL of Vero E6 cells were placed in 96-well tissue culture plates and incubated overnight at 37 °C. Serial dilution of the inoculated DMEM was performed and 200 µL of every dilution was re-used on the monolayers of Vero E6 cells. The cells were then incubated for one hour at 37 °C with shaking every 15 min. The inoculum was subsequently replaced with DMEM supplemented with 0.8% agarose and incubated for 4 days at 37 °C. The overlay was then removed and the cells were fixed for 15 min using 4% paraformaldehyde. SARS-CoV-2 titer was determined by counting plaques (plaque-forming units (pfu/mL), a measure used in virology to describe the number of virus particles capable of forming plaques per unit volume) after staining the cells with crystal violet. TCID50 for SARS-CoV-2 was calculated as 3.16 × 10^5^.

### 2.4. Immobilization of Lactoferrin on Q-Tips

Cotton is cellulose that contains polyhydroxy groups, and it was oxidized to an aldehyde functional group for the immobilization of lactoferrin (Figure 1-I,II). The cellulose Q-tips were activated by immersion in an acidic mixture of 100 mL of 2 mM sodium periodate (NaIO_4_) and 1 mL of sulfuric acid and left stirring at room temperature overnight. The next day, the Q-tips were thoroughly washed with deionized water to remove excess oxidizing agents. The oxidized Q-tips were characterized by FTIR spectroscopy. KBr pellets with the un-oxidized and oxidized Q-tips cotton were separated and run in the FTIR machine in the transmission mode. The results indicated the appearance of a new peak at 1730 cm^−1^ in the oxidized Q-tips, confirming the existence of an aldehyde group. Ultimately, the oxidized Q-tips were stored for further use.

The aldehyde functional groups on the Q-tips were used to conjugate the amino groups on the lactoferrin. The procedure was as follows: 50 µL of the lactoferrin was added to 1 mL of pH 7.4 PBS and mixed. The aldehyde functionalized Q-tips were immersed in buffered lactoferrin and left to stir overnight at 4 °C. The next day, the Q-tips were washed with PBS buffer to remove unbounded lactoferrin protein. BSA was used to block un-reacted aldehyde functional groups by incubating the Q-tips in a solution of 1 mg/mL BSA for an hour at room temperature. Then the Q-tips were washed extensively with PBS buffer and stored in PBS at 4 °C for further use. The control Q-tips were prepared by incubating the oxidized Q-tips in BSA solution (1 mg/mL) in PBS buffer at room temperature overnight. The next day, the BSA-conjugated Q-tips were extensively washed with PBS to remove non-bounded BSA. The BSA-conjugated control Q-tips were stored in PBS buffer at 4 °C for future use.

### 2.5. Immobilization of ACE2 Human Protein on the Orange Polymeric Nanoparticles

Orange polymeric nanoparticle suspension (400 µL, 2.5% aqueous suspension) was washed three times with 400 µL distilled water. Freshly prepared EDC/NHS solution was prepared by mixing 1 mg EDC and 1 mg NHS in 1 mL distilled water. A total of 500 µL of the freshly prepared EDC/NHS solution was incubated with the orange nanoparticles and mixed at room temperature for 20 min. Then, activated colored nanoparticles were washed five times with PBS buffer. Following, 30 µL of the ACE2 human protein was mixed with 200 µL PBS overnight at 4 °C. ACE2 human proteins conjugated to the orange nanoparticles were washed extensively using centrifuge at 5000 rpm using PBS buffer to remove the non-bonded ACE2 human protein. Finally, nanoparticles were incubated with 1 mg/mL BSA in PBS for one hour to block the active sites on the activated colored nanoparticles, followed by extensive washing with PBS to remove the bounded BSA. The ACE2 human protein conjugated to the nanoparticles was stored in PBS buffer at 4 °C for future use.

### 2.6. Sensing Protocol

The sensing protocol is summarized in Figure 1A,B. Lactoferrin was used as general virus-capturing agent and was conjugated to the Q-tips. The Q-tip-conjugated lactoferrin (Figure 1-III) was used to collect and pre-concentrate the samples from artificially contaminated surfaces (to mimic surfaces such as nose or throat) with a serial dilution of SARS-CoV-2. The number of the virus was titrated using plaque assay. The Q-tip-conjugated lactoferrin that captured the virus was washed thoroughly with PBS buffer to remove the non-captured virus particles from the Q-tips. Figure 1B illustrates the visualization and detection step of SARS-CoV-2 using lactoferrin-conjugated Q-tips following immersion in a solution called developing solution that contains orange nanoparticles conjugated with ACE2 human protein. Subsequently, the SARS-CoV-2 was sandwiched between the lactoferrin conjugated on the Q-tips and the ACE2 human protein conjugated with the orange nanoparticles. Control experiments were accomplished as described above, except the Q-tips swabbed the surfaces without SARS-CoV-2. Afterward, the Q-tips were immersed in the developing solution that contained the ACE2 human protein immobilized on the colored nanoparticles, followed by washing with PBS buffer solution to remove the unbound beads.

### 2.7. Quantitative Measurement of Q-Tip Color Change

The color change aligned with positive COVID-19 infection was detected visually and analyzed using an image analysis software (ImageJ) for quantitative detection. Color intensity was directly proportional to tested SARS-CoV-2 concentrations (10^3^ to 10^8^ pfu/mL). Following swabbing different SARS-CoV-2 solutions and subsequent placement in the developing solution containing the ACE2 conjugated with orange-colored nanoparticles, Q-tips were photographed and saved in JPEG format. The Q-tip images were imported into Image J software and the relative brightness of the pixels within the regions of interest was measured. To start the analysis, the measurement was set to generate the mean grey value and the uneven background was corrected by using the “Subtract Background” tool in the Image J program. Then, a defined, rectangular-shape macron was recorded within a specific-colored location in the Q-tip center and saved to be used in the analyses of all other photos. The mean grey area was inversely proportional to the color intensity, i.e., the mean grey area decreased by increasing test swab color intensity at higher SARS-CoV-2 concentrations. Quantitative measurement reliability was validated by calculating the average mean grey area of at least three regions of interest (defined shape) at different districts of the colored location.

### 2.8. Quantitative Method Validation

To evaluate SARS-CoV-2 diagnostic Q-tip quantitative ability and its correlation with a specific SARS-CoV-2 concentration, linearity was carried out on six different SARS-CoV-2 concentrations ranging from 10^3^–10^8^. The best fit line was obtained by linear regression analysis of the average mean grey area percentage against concentration in pfu/mL. Parameters such as the standard error of the response and the slope of the straight line were calculated.

## 3. Results and Discussion

To step up and fight the COVID-19 pandemic at a critical point, it is of interest to develop a portable, cheap, simple, rapid, and colorimetric biosensor suitable for on-site application. In fact, the ability to conduct a diagnostic test at home and the ability to read out the results with the naked eye will allow individuals to quarantine themselves immediately, thus preventing infection transmission and increasing treatment efficacy.

The present study demonstrates a colorimetric assay for the detection of the SARS-CoV-2 S1 protein as an identifying target and the ACE2 receptor as the recognition element. The colorimetric sensing biosensor comprises two parts (Q-tip detection platform immobilized with lactoferrin as general capturing element, and ACE 2 receptor labeled with orange-colored polymeric nanoparticles) that have different functions. Lactoferrin was primarily used because of its low cost, natural abundance, and ability to bind to most pathogens like viruses [27,28]. Moreover, Q-tips were used because they can safely collect and preconcentrate the viruses and are easily disposed of by incineration after use, making them more suitable for infectious disease testing. Moreover, Q-tips licenses their use as a disposable biosensor that can be mass-produced at low cost by companies or implemented by clinicians and regulatory agencies and update the public to better control the potential health risks. In addition, designed colorimetric biosensors minimize the time required to confirm positive cases between infection and symptom appearance, preferably covering the very early infection period (1–3 days) and eventually allowing virus replication to be monitored in asymptomatic patients.

In the Q-tip testing zone, lactoferrin was immobilized to capture incoming pathogens, including SARS-CoV-2, characterized by its S1 spike protein. For the diagnostic step, the colorimetric response was monitored visually following Q-tip testing zone pairing with the labeled ACE2 receptor. In this regard, color change from white to orange following decorated ACE2 pairing with the SARS-CoV-2 spike protein was observed comparatively with viral concentrations. A series of tests were performed in real nasopharyngeal samples (artificially infected with SARS-CoV-2 and compared with uninfected negative control samples), as shown in Figure 2A.

### 3.1. Quantification Measurement

For quantitative purposes, fabricated Q-tips were exposed to six different concentrations of SARS-CoV-2 (1.5 × 10^8^_,_ 1.5 × 10^7^, 1.5 × 10^6^, 1.5 × 10^5^, 1.5 × 10^4^, and 1.5 × 10^3^ pfu/mL), as presented in Figure 2B. The current study achieved a lower limit of detection (LOD) in the range of 100 pfu/mL in 5 min, demonstrating high sensitivity. The LOD was determined as the lowest SARS-CoV-2 concentration that could not reliably alter the white color of Q-tips (i.e., there was no visual color change). Negative control with no SARS-CoV-2 confirmed no visual color change (Figure 2). It is worth mentioning that the visual reading of the LOD was validated side by side with the control by two researchers in the lab. A minimum of three random respondents clarified the color change (white to orange).

An increase in orange color intensity of the Q-tip was directly proportional to SARS-CoV-2 concentration (1.5 × 10^3^_,_ 1.5 × 10^4^, 1.5 × 10^5^, 1.5 × 10^6^, 1.5 × 10^7^, 1.5 × 10^8^ pfu/mL), as illustrated in Figure 2. Initially, image brightness and contrast were assessed and normalized using ImageJ software (a public-domain, Java-based image-processing program developed at the National Institute of Health) [11,12,29,30,31,32]. Then, color intensity was quantified. Error bars represent the deviation of the colorimetric results from replications.

The calibration plot of different SARS-CoV-2 concentrations and the mean grey area percentage indicate a linear correlation. The linear part was fitted to a linear regression (y = 2.6857x + 21.6, R^2^ = 0.942) in the concentration range of 10^3^–10^8^ pfu/mL.

### 3.2. Cross-Reactivity Study

The specificity of fabricated SARS-CoV-2 sensing Q-tips was assessed by swapping different surfaces contaminated with other respiratory viruses such as Flu A, Flu B, and MERS COV. Afterward, Q-tips were incubated with the ACE2 receptor labeled with the orange-colored polymeric nanoparticles (Figure 3). SARS-CoV-2-sensing Q-tips did not reveal any color change in the presence of other viruses such as Flu A, Flu B, or MERS COV antigens on the contaminated surfaces, showing that non-specific adsorption on the sensor surfaces was insignificant.

### 3.3. Comparison with Alternative SARS-CoV-2 Biosensors Using S1 Protein as a Biomarker

Mavrikou et al. [33] developed a biosensor based on a bioelectric recognition approach for the detection of the SARS-CoV-2 S1 protein. The biosensor provides a result within 5 min with an LOD of about 1 pfu/mL and confirmed no cross-reactivity with the SARS-CoV-2 nucleocapsid protein. Another study introduced a one-step optical detection by applying a spike protein nano-plasmonic resonance biosensor. This biosensor detects virus particles (LOD = 370 pfu/mL) [34] in one step within 15 min. Another study conducted by Mahari et al. presented a gold-nanoparticle (NP)-based electrochemical biosensor for the detection of the SARS-CoV-2 spike S1 antigen as an identifying target. In this biosensor, gold NPs were utilized as a signal amplification for their significant electrical conductivity [35]. This immunosensor was sensitive with an LOD of about 10 pfu/mL. Another study that utilized a viral spike protein as an identifying target was based on the use of a cobalt-functionalized TiO_2_ nanotube-based electrochemical sensor. This sensor was specific, fast, and sensitive for detecting the S-receptor binding domain protein of SARS-CoV-2 at a low concentration of 14–1400 nM [36]. An innovative method for fast and sensitive detection of the SARS-CoV-2 S1 protein was established by utilizing the SARS-CoV-2 ACE2 receptor as an identifying element [36]. In a lateral flow immunoassay (LFIA), ACE2 and S1 protein-monoclonal antibodies were paired with each other for capture and detection.

### 3.4. Comparison with Alternative SARS-CoV-2 Colorimetric and Nano-Based Biosensors Using Other Biomarkers

Table 1 demonstrates a variety of up-to-date colorimetric and nano-based biosensing techniques presented for COVID-19 infection detection. Each assay is supported by its privilege of being highly sensitive and selective, yet their onsite practical use may be restricted by the pretreatment procedures required and large-scale availability. In brief, reverse transcription loop-mediated isothermal amplification (RT-LAMP) assays developed for SARS-CoV-2 detection [37,38,39] overcome the limitation of qRT-PCR-based assay. Another speedy and highly accurate assay (12 copies per reaction) was achieved by the application of a one-step RT-LAMP mediated with a nanoparticle-based biosensor (NBS) [37]; however, the assay duration time is 60 min [37]. On the other hand, a field-effect transistor (FET)-based biosensor in which the SARS-CoV-2 spike antibody is conjugated to a graphene sheet as a biosensing platform was highly sensitive and could detect 1 fg/mL of SARS-CoV-2 spike protein in PBS and 100 fg/mL in the clinical transport medium [40]. Yet, the resulting readout is based on instrumentation. Recently, Ren et al. reported the development of a paper-based magnetic-focus-enhanced lateral flow assay for the detection of SARS-CoV-2. In this lateral flow assay, horseradish peroxidase was used to catalyze the colorimetric reaction for the amplification of the colorimetric signal. The assay depicted an LOD of 400 PFU/mL in PBS buffer and 1200 PFU/mL in saliva samples with 66.7% sensitivity and 100% specificity. However, several washing steps are required during assay preparation.

Interestingly, the current diagnostic probe presents an affordable and rapid bioassay that can diagnose samples within 5 min of sample/sensor contact with high sensitivity and specificity.

## 4. Conclusions

This study involved a major drive to develop a novel technology that provides low-cost in-situ testing to facilitate COVID-19 infection detection and consequent treatment, affording implementation in real-life situations, saving time, and enabling the correct action to be taken on time with minimal interventions. The developed assay generates a visual signal at the Q-tip testing zone from white to orange upon pairing the Q-tip infected with SARS-CoV 2 with an orange polymeric nanoparticle carrying ACE2 receptor. The biosensor lower visual detection limit of SARS-CoV 2 was 100 pfu/mL within 5 min and the linear range of quantitation was 10^3^–10^8^ pfu/mL. This integrated approach will result in low-cost, rapid, and accurate detection of SARS-CoV 2 in situ with immediate targeted treatment, eliminating the need for any sample processing. Besides, this technology is not limited to COVID-19 infection only, but could also be applied to other emerging viral infections and pathogens.

## Figures and Tables

**Figure 1 biosensors-12-00232-f001:**
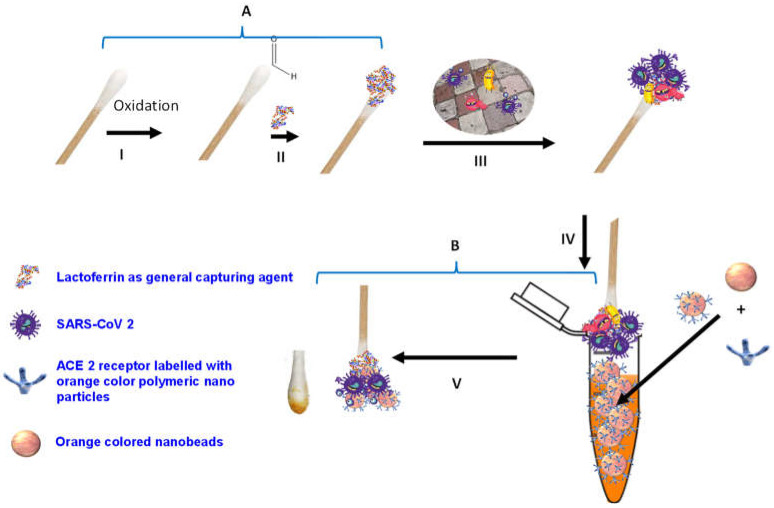
Schematic diagram of SARS-CoV-2 colorimetric sensing Q-tips. (**A**) Preparation of cotton swab-lactoferrin sensor. (**I**) Oxidation activation of the Q-tip cotton swab; **(II)** immobilization of lactoferrin capturing agent on cotton swab surface; (**III**) capturing of SARS-CoV-2. (**B**) Colorimetric detection assay (**IV**) of SARS-CoV-2 spike protein sandwiched between lactoferrin immobilized over cotton swab surface and ACE2 receptor decorated with the orange polymeric nanoparticle; (**V**) SARS-CoV-2 colorimetric detection.

**Figure 2 biosensors-12-00232-f002:**
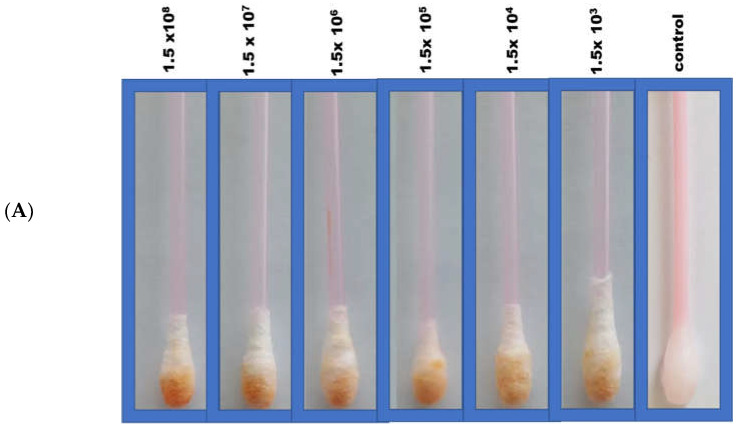
(**A**) Colorimetric detection of different SARS-CoV-2 concentrations (10^3^–10^8^ pfu/mL). (**B**) Quantitative load of the SARS-CoV-2 collected by the swab and the calibration curves of the different SARS-CoV-2 concentrations using the sandwich colorimetric immunoassay; a plot of the colored area percentage versus SARS-CoV-2 concentrations.

**Figure 3 biosensors-12-00232-f003:**
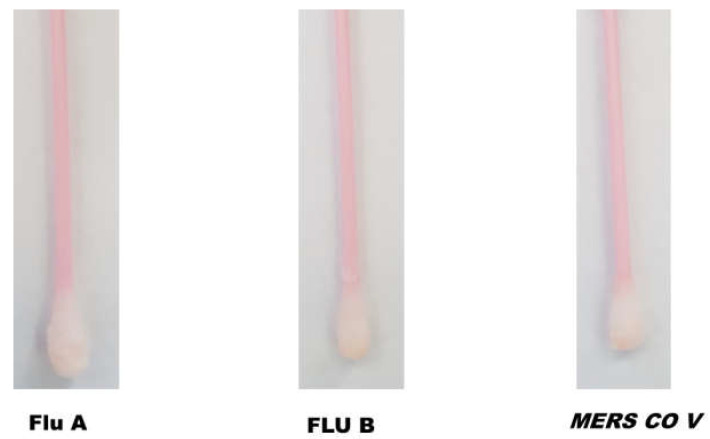
Cross-reactivity of SARS-CoV-2-fabricated Q-tips against other coronaviruses, including Flu A, Flu B, and MERS COV.

**Table 1 biosensors-12-00232-t001:** Shows a variety of up-to-date colorimetric and nano-based biosensing techniques presented for COVID-19 infection detection.

Technique	Method	Target	Sample	Time (min)	Sensitivity and Specificity (%)	Refs.
**Colorimetric**	RT-LAMP	Viral RNA	Saliva	45	82.6 and 100%	[41]
	RT-LAMP	Viral RNA	Throat	30	High sensitivity	[42]
	RT-LAMP	Viral RNA	Saliva	30	85 and 100%	[43]
	RT-LAMP	Primer set targets orf7a, orf7b, and orf1ab regions of SARS-CoV-2.	Saliva	60	97 and 100%	[44]
	RT-PCR DNAzyme-based sensor	Viral RNA	Nasopharyngeal		100 and 100%	[45]
	Loop-mediated isothermal amplification	Primer set targets ORF1ab and nucleocapsid (N) genes of SARS-CoV-2	Saliva	16	High sensitivity	[46]
**Fluorometric**	RT-PCR		Saliva		92.7 and 99.9%	[47]
	RT-PCR	RNA polymerase (RdRp)/helicase (Hel), spike (S), and nucleocapsid (N) genes of SARS-CoV-2	Nasopharyngeal, throat, and sputum	Long processing time	High sensitivity and specificity	[48]
**Nano-based biosensors**	Dual-functional plasmonic biosensor	RdRp-COVID, ORF1ab-COVID, and E genes from SARS-CoV-2	Viral RNA		Highly sensitive	[49]
	Reverse transcription loop-mediated isothermal amplification combined with nanoparticle-based biosensor	Primer set targets of F1ab and nucleoprotein genes of SARS-CoV-2	Oropharynx swab	60	High sensitivity and specificity	[50]
	Field-effect transistor based biosensor	Specific antibody against SARS-CoV-2 spike protein	Nasopharyngeal		Highly sensitive	[40]
	Magnetic-focus-enhanced lateral flow assay	Specific antibody against SARS-CoV-2 spike and nucleocapsid protein	Buffer, saliva, nasal swab		66.7% sensitivity and 100% specificity	[7]

## Data Availability

Not applicable.

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
