# Peer review of "A Portable Nanoprobe for Rapid and Sensitive Detection of SARS-CoV-2 S1 Protein"

_biosensors, 2022, doi:10.3390/bios12040232_

Round 1

Author Response

attached the reply to reviewer 1

Reviewer 2 Report

Title: Rapid Nanoprobe Biosensor for the Detection of SARS‐CoV‐2

Journal: Biosensors

Recommendation: In this study authors developed a Nanoprobe Biosensor for the Detection of SARS‐CoV‐2 in clinical samples”. Developed Nanoprobe Biosensor was tested for various real sample analysis. I think that paper is interesting based on the analytical aspects and suggest for publication in this high-quality journal after below corrections (Major revision):

  • Title Should be revise more effectively.
  • Keywords:Note that an article will be found more often by search engines if many significant keywords are provided. Authors must change these keywords “onsite detection; low-cost diagnostic assay”
  • The real sample experiment with the procedure should be improved.
  • Introduction section must be revise more effectively and follow more suitable research articles.
  • Results: Some experimental data lack (relative) standard deviations. Give averaged data for important experimental data along with standard deviations (±) and the number of experiments (n =?).
  • Figure quality need to improve.
  • General Comment: There are many grammatical and typographical errors. Please check the manuscript and refine carefully.
  • General: Authors should check all figures and captions in the manuscript.
  • Authors must improve the selectivity of the detection. Its important for selective detection in analytical aspects.
  • Figure 1. The schematic diagram is not clear. Authors must check.

Author Response

attached

This manuscript is a resubmission of an earlier submission. The following is a list of the peer review reports and author responses from that submission.

Round 1

Reviewer 1 Report

this paper presents a simple method to detect SARS-CoV2 with a rapid and low cost tool. It is a colorimetric sensor able to sandwich COVID 2 virus spike protein between the lactoferrin general capturing agent and the  complementary ACE2-labeled receptor. ACE2 receptor is decorated with an orange-colored polymeric nanoparticle to generate an optical visual signal upon pairing with the COVID 19 spike protein. This colorimetric change  from white to orange confirms a positive result. The visual detection limit of the COVID 19 engineered colorimetric sensor was 100 pfu mL-1 within a relatively short turnaround time of 5 min and the linear working range of quantitation was 10^3 - 10^8 pfu/mL. 

The idea to use colorimetric sensors to detect SARS-CoV2 is not new and has been explored in a large number of papers. Therefore this paper could be considered as a technical paper similar to the state-of-the-art.

The paper reports several errors and imprecisions. First of all, COVID-19 is not the name of the virus!! but the name used to identify the disease due to the virus! without considering that along the manuscript we can read COVID or CVOID this is in all the case an conceptual and technical important error!

At line 49-51 the authors claim that there is an urgency to develop rapid and reliable tests. Actually there is an interest to develop these tests, but not an urgency! SARS-CoV2 can be detect with multiple low cost rapid tests now (actually several million of them are done every day!)

Anyway, the idea to use a simple colorimetric in-tube reaction can be interesting, but from the results reported here it does not seem that this method has a good sensitivity. Figure 2 reports a linear trend, but looking at the error bars there is no clear difference between the different high concentration tested.

The authors should discuss the meaning to detect 1500pfu/mL and compare this number with the sensitivity of standard rapid tests (moreover a discussion on the viral concentration in real cases should be useful)

COVID-19 is a topic that has been extremelly hot during the last 2 years and there are tons of papers on that...the authors report here just 25 refs (with 6 auto-references).

An extensive integration with additional discussion on the state-of-the-art is a fundamental step to consider this paper for publication (considering the low level of novelty)

some important papers on colorimetric assay that should be mentioned are:

ACS Sens. 2020, 5, 10, 3043–3048

doi: 10.1126/scitranslmed.abc7075.

doi.org/10.1016/j.biosx.2021.100076

doi.org/10.3389/fmolb.2020.586254

doi.org/10.1101/2020.05.05.20092288

doi.org/10.3390/ijms21155380

doi.org/10.2144/btn-2020-0159

Papers on the topic of SARS-CoV2 detection by means of nanomaterials or engineered tools:

doi.org/10.1039/D0MA00702A

doi.org/10.1021/acsomega.1c04012

doi.org/10.1021/acssensors.1c00312

doi: 10.1007/s13205-020-02369-0

doi: 10.1016/j.lfs.2021.119117

doi.org/10.1038/s41587-021-00878-8

Author Response

Reviewer 1:

Comments and Suggestions for Authors

this paper presents a simple method to detect SARS-CoV2 with a rapid and low cost tool. It is a colorimetric sensor able to sandwich COVID 2 virus spike protein between the lactoferrin general capturing agent and the  complementary ACE2-labeled receptor. ACE2 receptor is decorated with an orange-colored polymeric nanoparticle to generate an optical visual signal upon pairing with the COVID 19 spike protein. This colorimetric change  from white to orange confirms a positive result. The visual detection limit of the COVID 19 engineered colorimetric sensor was 100 pfu mL-1 within a relatively short turnaround time of 5 min and the linear working range of quantitation was 10^3 - 10^8 pfu/mL. 

The idea to use colorimetric sensors to detect SARS-CoV2 is not new and has been explored in a large number of papers. Therefore, this paper could be considered as a technical paper similar to the state-of-the-art.

1-The paper reports several errors and imprecisions. First of all, COVID-19 is not the name of the virus!! but the name used to identify the disease due to the virus! without considering that along the manuscript we can read COVID or CVOID this is in all the case an conceptual and technical important error!

Thanks to the reviewer for this important notification. COVID-19 was only used in the revised version to refer to the infection and SARS-CoV-2 for virsus detection.

2- At line 49-51 the authors claim that there is an urgency to develop rapid and reliable tests. Actually there is an interest to develop these tests, but not an urgency! SARS-CoV2 can be detect with multiple low cost rapid tests now (actually several million of them are done every day!).

Thanks to the reviewer for the comment.  SARS-CoV-2 detection by molecular testing is based on samples sent to the lab that might be at risk of contamination during transportation and/or processing steps.  Accordingly, optimum sample collection, transportation, and processing technique are crucial. Interestingly, Q-tips swabs present a simple, cheap and convenient collection tool being widely available in all clinical ward areas. A new section in the revised version was added to compare the current diagnostic method with other colorimetric and nanobased techniques and present the difference in some merit figures as the sample running time and the lower limt of detection.

(3.3. Comparison with alternative SARS-CoV-2  colorimetric and nano-based biosensors using other biomarkers

Table 1 demonstrates a variety of up to date colorimetric and nanobased biosensing techniques presented for COVID-19 infection detection. Each assay is supported by its privilege being highly sensitive and selective, yet their onsite practical use may be restricted by the pretreatment proceduces required and large-scale availability. In brief, reverse transcription loop-mediated isothermal amplification (RT-LAMP) assays developed for SARS-CoV-2 detections27-29 overcomes the limitation of qRT-PCR based assay. Other speedy and highly accurat (12 copies per reaction) assay was achieved by the application of a one-step RT-LAMP mediated with nanoparticles-based biosensor (NBS)27, However, assay duration time is 60 min27.  On the other hand, field-effect transistor (FET) based biosensor in which the SARS-CoV-2 spike antibody is conjugated to a graphene sheet as a biosensing platform was higly sensitive and could detect 1 fg/mL of SARS-CoV-2 spike protein in phosphate-buffered saline and 100 fg/mL in the clinical transport medium30. Yet, result read out is based on instrumentation. Interstingly, the current diagnostic probe presents affordable and rapid bioassay which can diagnose samples within 5 min of sample/sensor contact with high sensitivity and specicificty )

3-Anyway, the idea to use a simple colorimetric in-tube reaction can be interesting, but from the results reported here it does not seem that this method has a good sensitivity. Figure 2 reports a linear trend, but looking at the error bars there is no clear difference between the different high concentration tested.

Thanks to the reviewer comments. Figure 2 was amended, and error bars were added. Also, a new section was added to the revised version.

(2.8 Quantitative method validation

To evaluate SARS-CoV-2 diagnostic Q-tips quantitative ability and its correlation with a specific SARS-CoV-2   concentration. Linearity was carried on six different SARS-CoV-2 concentration ranging from 103-108. The best fit line was obtained by linear regression analysis of the average mean grey area percentage against concentration in pfu/ml. Parameters such as the standard error of the response and the slope of the straight line were calculated. )

4- The authors should discuss the meaning to detect 1500pfu/mL and compare this number with the sensitivity of standard rapid tests (moreover a discussion on the viral concentration in real cases should be useful).COVID-19 is a topic that has been extremelly hot during the last 2 years and there are tons of papers on that...the authors report here just 25 refs (with 6 auto-references).An extensive integration with additional discussion on the state-of-the-art is a fundamental step to consider this paper for publication (considering the low level of novelty) some important papers on colorimetric assay that should be mentioned are:

 Response:

Thanks for the reviewer comments. A new section (3.3 Comparison with alternative SARS-CoV-2  colorimetric and nano-based biosensors using other biomarkers) and a new table (Table 1) was added to the revised version to cover reviewer concerns. This table pinpoints the current biosensing probe superiority.

Reviewer 2 Report

In this manuscript, the authors reported a rapid colorimetric biosensor for detecting SARS-COV-2 by sandwiching COVID 19 virus spike protein between the lactoferrin general capturing agent and the ACE2-labeled receptor. This sensor owns a visual detection limit of 100 pfu/ML within 5 min, providing a low cost in-situ testing approach to facilitate covid 19 detection and consequent treatment. However, I have several concerns before this manuscript can be accepted. Therefore, in its current form, revisions are needed.

1.The scale bar should be given in Figure 2 and 3.

  1. Since color intensity is directly proportional to COVID 19 concentration, how did the authors determine the targeted region (the whole Q-tip or a fixed region) to calculate the mean intensity? The sizes of those Q-tips are different in Figure 2 and will this influence the detection results?

  1. Did the authors conduct the optimization of the operating process, such as the concentration of each agent, the incubation time?

  1. Will it be possible to realize a lower detection limit of COVID 19 using this sensor? Could the author make some discussions?

Author Response

Reviewer 2:

Comments and Suggestions for Authors

In this manuscript, the authors reported a rapid colorimetric biosensor for detecting SARS-COV-2 by sandwiching COVID 19 virus spike protein between the lactoferrin general capturing agent and the ACE2-labeled receptor. This sensor owns a visual detection limit of 100 pfu/ML within 5 min, providing a low cost in-situ testing approach to facilitate covid 19 detection and consequent treatment. However, I have several concerns before this manuscript can be accepted. Therefore, in its current form, revisions are needed.

1.The scale bar should be given in Figure 2 and 3.

 Thanks to the reviewer for the comment. Figure 2 was amended.

2.Since color intensity is directly proportional to COVID 19 concentration, how did the authors determine the targeted region (the whole Q-tip or a fixed region) to calculate the mean intensity? The sizes of those Q-tips are different in Figure 2 and will this influence the detection results?

Thanks to the reviewer for the interesting comment. A new section in the experimental part was added to clarify this point.

2.7. Quantitative measurement of Q-tip colour change

The color change aligned with positive COVID-19 infection was detected visually and analysed using an image analysis software (ImageJ) for the purpose of quantitative detection. Color intensity was directly proportional to tested SARS-CoV-2 concentrations (103 to 108 pfu/ml). Following the swabing of different SARS-CoV-2 solution and subsequent immersement in the developing solution containing the ACE2 conjugated with orange colored nanparticles, Q-tips were photographed and saved in JPEG format. Q-tips images were imported into image J software and the relative brightness of the pixels within the regions of interest was measured.  To start analysis, measurement was set to generate the mean grey value and the uneven background was corrected by using the ˝Subtract  Background˝ tool in image J program. Then, a defined, rectangular shape macron was recorded within a specific colored location in the Q-tip centre and saved to be used in the analyses of all other photos. The mean grey area was inversely proportional to the color intensity, i.e., the mean grey area decreases by increasing test swab color intensity at higher SARS-CoV-2  concentrations. Quantitative measurement reliability was validated by calculating the average mean grey area of at least three regions of interest (defined shape) at different districts of the colored location.

2.8 Quantitative method validation

To evaluate SARS-CoV-2 diagnostic Q-tips quantitative ability and its correlation with a specific SARS-CoV-2   concentration. Linearity was carried on six different SARS-CoV-2 concentration ranging from 103-108. The best fit line was obtained by linear regression analysis of the average mean grey area percentage against concentration in pfu/ml. Parameters such as the standard error of the response and the slope of the straight line were calculated.

  1. Did the authors conduct the optimization of the operating process, such as the concentration of each agent, the incubation time?

 Thanks for the reviewer interesting comment. All operating process were optimized in advanced and the optimum parameters are presented in the current study.

  1. Will it be possible to realize a lower detection limit of COVID 19 using this sensor? Could the author make some discussions?

Thanks for the reviewer critical comment. In each trial, a negative control sample must be tested to validate color change. And that’s why we have mentioned in the result and discussion section that a comparison of color change was done side-by-side with the control to clarify orange color difference. A minimum of two researchers had visually clarified the lowest concentration resulted in a noticeable or distinguishable orange color compared to the white color control.

Round 2

Reviewer 1 Report

In my previour report I tryied to report several criticisms in the paper in order to help the authors to improve it and to enable the publication.

Actually, I am not happy at all with the reply and the modification in the text.

As reported: 

My previous comment:

 At line 49-51 the authors claim that there is an urgency to develop rapid and reliable tests. Actually there is an interest to develop these tests, but not an urgency! SARS-CoV2 can be detect with multiple low cost rapid tests now (actually several million of them are done every day!).

Reply:

Thanks to the reviewer for the comment.  SARS-CoV-2 detection by molecular testing is based on samples sent to the lab that might be at risk of contamination during transportation and/or processing steps.  Accordingly, optimum sample collection, transportation, and processing technique are crucial. Interestingly, Q-tips swabs present a simple, cheap and convenient collection tool being widely available in all clinical ward areas. A new section in the revised version was added to compare the current diagnostic method with other colorimetric and nanobased techniques and present the difference in some merit figures as the sample running time and the lower limt of detection.

THIS IS NOT CORRECT! at the moment the rapid detection of SARS-CoV-2 by means of rapid tests is an established procedure and no sample transportation or other critical steps are reported by the authors are needed! This is true for molecular tests, but it is a completely different story (the authors are not presenting something that can be comparad to molecular tests)

MY COMMENT:

Anyway, the idea to use a simple colorimetric in-tube reaction can be interesting, but from the results reported here it does not seem that this method has a good sensitivity. Figure 2 reports a linear trend, but looking at the error bars there is no clear difference between the different high concentration tested.

REPLY:

Thanks to the reviewer comments. Figure 2 was amended, and error bars were added. Also, a new section was added to the revised version.

the error bars is figure 2 clearly show that there is not significant difference between the measurements performed at different concentrations. Also the figure shows that it is really hard to appreciate the different concentrations tested! this limit significantly the use of this method

Finally, I recommended to better introduce the topic while the authors just reported some comparison with other papers on colorimetric tests (completely missing the introduction on SARS-CoV-2 detection). also the comparison is wrong because they compare the sensitivity using different units!!

I think that this paper is not suitable for publication in Biosensors (also considering the good impact of the journal)

Author Response

plz find attached file

Reviewer 2 Report

Authors have responded the reviewer comments in detail and I find this paper complete after this revision.

Author Response

plz see attached file

Round 3

Reviewer 1 Report

I have already rejected this manuscript during the previoust step. I don't see any particular improvement that could justfy another decision